# Molecular Pathophysiology of Parathyroid Tumorigenesis—The Lesson from a Rare Disease: The “MEN1 Model”

**DOI:** 10.3390/ijms252111586

**Published:** 2024-10-29

**Authors:** Alessandro Brunetti, Roberta Cosso, Fabio Vescini, Alberto Falchetti

**Affiliations:** 1SC Endocrinologia, Ospedale Santa Maria della Misericordia, 33100 Udine, Italy; alessandro.brunetti94@gmail.com (A.B.); fvescini@alice.it (F.V.); 2Medical Center Amedic, 47521 Cesena, Italy; robcos63@libero.it; 3SC Endocrinologia, ASST Grande Ospedale Metropolitano Niguarda, Piazza dell’Ospedale Maggiore, 3, 20162 Milano, Italy

**Keywords:** primary hyperparathyroidism, familial hyperparathyroidism, genetics of parathyroid tumors, MEN1 syndrome, MEN4 syndrome, MEN1 gene, menin, CDKN1B gene, miRNAs, epigenetics

## Abstract

Primary hyperparathyroidism represents the third most prevalent endocrine disease in the general population, consisting of an excessive secretion of parathyroid hormone from one or, more frequently, more of the parathyroid glands, leading to a dysregulation of calcium homeostasis. Schematically, its development occurs primarily by pathophysiological events with genetic mutation, at the germline and/or somatic level, that favor the neoplastic transformation of parathyroid cells and promote their aberrant proliferation, and mutations determining the shift in the PTH “set-point”, thus interfering with the normal pathways of PTH secretion and leading to a “resetting” of Ca^2+^-dependent PTH secretion or to a secretion of PTH insensitive to changes in extracellular Ca^2+^ levels. Familial syndromic and non-syndromic forms of primary hyperparathyroidism are responsible for approximately 2–5% of primary hyperparathyroidism cases and most of them are inherited forms. The history of the genetic/molecular studies of parathyroid tumorigenesis associated with multiple endocrine neoplasia type 1 syndrome (MEN1) represents an interesting model to understand genetic–epigenetic–molecular aspects underlying the pathophysiology of primary hyperparathyroidism, both in relation to syndromic and non-syndromic forms. This minireview aims to take a quick and simplified look at the MEN1-associated parathyroid tumorigenesis, focusing on the molecular underlying mechanisms. Clinical, epidemiological, and observational studies, as well as specific guidelines, molecular genetics studies, and reviews, have been considered. Only studies submitted to PubMed in the English language were included, without time constraints.

## 1. Introduction

Primary hyperparathyroidism (PHPT) is distinguished by an excessive secretion of parathyroid hormone (PTH) from one or more of the parathyroid glands, leading to a dysregulation of calcium homeostasis. PHPT is one of the most prevalent endocrine disorders in the general population, with an estimated frequency of 3 cases per 1000 people in Europe. This prevalence increases to 21 cases per 1000 in post-menopausal women [1]. Regardless of its genetic or molecular characteristics, the development of PHPT is primarily the result of two factors: (a) a genetic mutation, whether germline or somatic, that enables the neoplastic transformation of parathyroid tissue, resulting in an aberrant proliferation of parathyroid cells and an increase in parathyroid cell mass; (b) a shift in the PTH “set-point” caused by mutations that interfere with the normal pathways of PTH secretion, leading to a “resetting” of Ca^2+^-dependent PTH secretion or to a secretion of PTH that is insensitive to changes in extracellular Ca^2+^ levels [2,3] (Figure 1).

The development of PHPT is influenced by a complex interplay of genetic, molecular, and environmental factors, representing a challenge for both researchers and clinicians. When the delicate equilibrium of parathyroid function is disrupted, the stage is set for the emergence of parathyroid tumors, most notably adenomas and only rarely carcinomas, precipitating the cascade of events leading to PHPT. While most cases of PHPT arise sporadically, a subset exhibits familial clustering. Familial PHPT (FPHPT) can be classified into two groups: syndromic forms, where PHPT is part of a complex clinical picture involving several other organs, and non-syndromic forms, where PHPT occurs as an isolated disorder. Familial syndromic and non-syndromic forms of PHPT are responsible for approximately 2–5% of PHPT cases [2,3] (Table 1). A further distinction within FPHPT should be made between inherited and heritable forms. This difference is subtle and can be summarized as follows: (a) the inherited form is when the known causal germinal genetic alteration, generally heterozygous, has been transmitted to offspring; (b) a heritable form indicates the existence of an identified de novo germline mutation in a patient with PHPT, without suspicion/evidence of familial cases but potentially transmissible to offspring, triggering a future FPHPT.

The investigation of FPHPT yielded numerous insights into the genetic pathways that contribute to parathyroid tumorigenesis, including modifications to critical genes such as *CDC73*, *MEN1*, and *CASR*. Each of these genes orchestrates a unique role in the dysregulation of cell cycle control, apoptosis, and calcium sensing [2,3]. Unfortunately, no clear genotype–phenotype correlation is observed in PHPT. Indeed, a single genetic mutation can result in a variety of parathyroid pathological conditions, including adenomas, diffuse hyperplasia, and, in rare cases, carcinomas. Additionally, the clinical presentation of these conditions is frequently overlapping, making it challenging to differentiate malignant and benign disease at a pre-operative level. Even after surgery is performed, no specific histological characteristics allow one to define the parathyroid carcinoma [4].

Therefore, it is vital to increase our insight on genetic, epigenetic, and molecular signatures driving parathyroid tumorigenesis and to identify novel diagnostic biomarkers that can distinguish the different kinds of parathyroid neoplasms, as well as to develop innovative therapeutic strategies. The study of parathyroid tumorigenesis associated with multiple endocrine neoplasia type 1 syndrome (MEN1) represents an excellent paradigm for understanding genetic–epigenetic–molecular aspects underlying the pathophysiology of PHPT, both in relation to the syndromic and non-syndromic forms.

This minireview aims to take a quick look at the history of parathyroid tumorigenesis associated with MEN1, focusing on the molecular underlying mechanisms. Clinical, epidemiological, and observational studies, as well as specific guidelines, molecular genetics studies, and reviews, have been considered. Only studies submitted to PubMed in the English language were included, without time constraints.

The following keywords have been used: parathyroid tumorigenesis; parathyroid tumors; multiple endocrine neoplasia type 1 syndrome (MEN1); MEN1 gene; allelic loss; loss of heterozygosity; MEN1 parathyroid tumors; tumor suppressor genes; oncogenes; epigenetics; oncomir; microRNAs/miRNAs.

## 2. Multiple Endocrine Neoplasia Type 1 (MEN1) Syndrome and *MEN1* Gene

MEN1 syndrome (OMIM#131100) is a highly penetrating rare autosomal dominant inherited tumor syndrome, typically characterized by the presence of parathyroid, pituitary, and gastro–entero–pancreatic (GEP) neuroendocrine tumors (NETs) [5,6]. A combination of other endocrine and non-endocrine tumors may also be present, such as thymic, bronchial carcinoids, adrenocortical tumors, facial angiofibromas, lipomas, and collagenomas [7,8,9], resulting in a broad spectrum of phenotypic presentations. The majority of patients with the inherited form of MEN1 hosts germline mutations of the *MEN1* gene [10]. Several germline mutations of the *MEN1* gene have been described to date, both in inherited and de novo MEN1 cases [11,12]. These mutations are mainly heterozygous inactivating mutations located in the coding region of the *MEN1* gene [13,14,15]. Specifically, around 10–15% are nonsense mutations, 40% are frameshift insertions and deletions, and 25% are missense mutations, while approximately 11% are splice site defects [3].

## 3. MEN1-PHPT

The parathyroid tumors linked to MEN1 syndrome typically exhibit benign growth and affect the whole parathyroid tissue. In particular, an asynchronous and asymmetric hyperplasia of all parathyroid glands is observed, explaining why the multiglandular parathyroid involvement may be missed at the time of initial surgery and be confused with a single adenoma [16]. (Figure 2). Fewer than 20 cases of parathyroid carcinoma (PC) have been published to date in MEN1 patients [17]. Although extremely rare in general, the presence of PC is always possible and should be considered in patients with symptomatic FPHPT and PHPT.

## 4. Epidemiological Aspects

In line with an autosomal dominant transmission, an equal distribution between the sexes is observed. Frequently, PHPT is the first biochemical–clinical presentation of the syndrome, since the onset of PHPT in MEN1 typically occurs during the second decade of life [2,3]. Penetrance reaches approximately 50% within 20 years of age, and 100% within 50 years, occurring approximately 3 decades in advance with respect to sporadic PHPT in the general population [5]. In an observational study on 160 MEN1 patients, PHPT was present in 58% of subjects aged 15–20 years, in 32% between 10–15 years, and in 10% during the first decade of life, including also reports on PHPT in a 4-year-old child [18].

## 5. Clinical Presentation

Primary hyperparathyroidism in MEN1 (MEN1-PHPT) is distinguished by early onset and clinical aggressiveness. Rapid loss of bone mass is observed, particularly at the cortical level as in classic PHPT, but with a more severe bone impairment and increased risk for fragility fractures than sporadic forms [19,20]. Up to 50% of patients with MEN1 present reduced bone mineral density (BMD) before 50 years of age [19,21]. Moreover, recurrent nephrolithiasis is one of the most frequent clinical manifestations of MEN1, frequently occurring before 30 years of age. Up to two-thirds of patients with MEN1 present complications of urolithiasis (renal/ureteric colic, urinary tract infection) as the first clinical manifestation of the syndrome, and, in half of cases, urolithiasis is reported as the only clinical manifestation [22,23]. Notably, PTH and calcium elevation in MEN1-PHPT is often milder than in sporadic PHPT, especially in younger patients. Indeed, up to one-third of patients <50 years with MEN1-PHPT actually exhibit PTH levels within the normal range [19]. In this regard, it has been suggested that an MEN1 patient aged <50 years with normal PTH levels is 13.5 times more likely to develop PHPT than a non-MEN1 patient aged >50 years with elevated PTH levels [2,4,24].

A comprehensive evaluation of complications associated to PHPT should therefore be performed in all MEN1 patients, regardless of PTH levels. Furthermore, genetic screening for MEN1 should be offered to patients with familial cases of PHPT, with early-onset hyperparathyroidism and those with multi-gland hyperparathyroidism, regardless of age [24].

## 6. MEN1 Gene

The *MEN1* gene (NM_130799.2) is a tumor suppressor gene located on chromosome 11q13 spanning approximately 9000 bp of genomic DNA and organized in 10 exons transcribed into a 2.8 kb mRNA, with the translational start codon located in exon-2 and the stop codon in exon-10 [10,16]. Its encoded product is the menin protein, of which different isoforms have been reported: a long isoform consisting of 1615 amino acids, the canonical sequence, a short isoform of 2610 amino acids in length, and another isoform consisting of 575 amino acids [11]. Menin has three nuclear localization signals (NLSs), located in its C-terminal region, at codons 479–497 (NLS1), 546–572 (NLSa), and 588–608 (NLS2), respectively, and five putative guanosine triphosphatase (GTPase) sites (G1–G5). Such NLSs directly interact, in a sequence-independent manner, with DNA as a scaffold protein controlling gene expression and cell signaling. Different menin regions have been implicated in the binding to different interacting proteins, such as JunD, GFAP, NFκB, Smad1/5, Pam, NM23H1, RPA2, Runx2, MLL, NMMHC II-A, FANC2, mSin3A, HDAC1, Ask, vimentin, CHES1, and estrogen receptor-alpha (ER-alpha). Their functions, within the menin’s partnership, may vary from genomic stability control (RPA2, FANC2) to cell division (vimentin, NMMHC II-A, GFAP), cell cycle control (NM23, ASK), gene transcription regulation (JunD, NFκB, Smad proteins, Runx2, MLL, ER-alpha, CHES1), and epigenetic regulation (MLL, HDAC).

In addition, menin has an important role within the selective mediation of chromatin remodeling and, consequently, in both gene expression and cell proliferation regulation [10]. The disruption and/or alteration of one or more of these molecular partnerships may account for tumor susceptibility, including the parathyroid tissue.

## 7. *MEN1* Gene’s Role in Parathyroid Tumorigenesis

Parathyroid tumorigenesis in MEN1 results from the combination of a germline *MEN1* mutation together with a somatic mutation occurring in a parathyroid tumor precursor cell. Typically, loss of heterozygosity (LOH) for polymorphic DNA markers is found at the chromosome 11q13 region, where the *MEN1* gene has been mapped (Figure 3). This genetic feature aligns with the expected behavior of a tumor suppressor gene (TSG) [11,25,26]. The time interval, probably variable and multifactorial, between the occurrence of the second mutational event in the parathyroid tissue and the onset of clinical or biochemical manifestation of PHPT is unknown [10].

Interestingly, whole-exome studies identified *MEN1* mutation/inactivation as a common molecular event also in sporadic parathyroid adenomas, unraveling the occurrence of somatic *MEN1* mutations in about 35–40% of investigated sporadic tumors [27,28,29]. Truncating mutations of the *MEN1* gene are the most frequent, together with LOH at the 11q13 locus, which has been reported in about 40% of sporadic parathyroid adenomas [30,31].

## 8. Molecular Genetic Studies of MEN1-PHPT Tissues

They can be “scholastically” divided into before and after the cloning of the *MEN1* gene.

### 8.1. Before the MEN1 Gene Cloning

Before the identification of the *MEN1* gene in the late 1990s, parathyroid tumors in MEN1 were considered as the result of a polyclonal expansion [32,33,34]. A contribution to the current genetic approach to parathyroid tumorigenesis in MEN1 was provided by a study involving patients affected by MEN1-PHPT, in whom DNA extracted from parathyroid tissues was compared with the constitutive DNA extracted from whole blood [35]. The results of this study suggested that MEN1-hyperplastic parathyroid tumors could be regarded as monoclonal lesions, progressing or beginning by the germline inactivation of the *MEN1* gene, followed by somatic loss of the wild copy of the gene. Lately, clonal composition analyses of micro-dissected sections from abnormal MEN1 parathyroid tissue on DNA obtained from single nodules and non-nodular areas, used by both chromosome 11q13 microsatellite PCR and patterns of X chromosome inactivation, highlighted a genetic heterogeneity in MEN1 parathyroid microareas, exhibiting a “clonal” pattern for allelic losses [36,37]. Specifically, the cytohistological structure did not result in being useful predictors of clonality but these findings strongly suggested that, in familial MEN1 parathyroid tumors, the nodular pattern growth, large size of chief cells and nuclear atypia were associated with monoclonality. Interestingly, the LOH at the 11q13 region is observed in the majority of MEN1 parathyroid tumors, while it is less frequent in the sporadic counterpart, confirming the role of an inactivation of an 11q13-located tumor suppressor gene (TSG) in parathyroid tumorigenesis, but also highlighting an apparent heterogeneity within sporadic parathyroid tumors [38]. These findings may be considered prodromal to the more recent redefinition of clonality of parathyroid hyperplasia reported in the 2022 WHO guidelines, endorsing PHPT-related multiglandular parathyroid disease as a germline susceptibility-driven neoplasm [39].

### 8.2. After the MEN1 Gene Cloning

At the Catalogue of Somatic Mutations in Cancer (COSMIC), only a subset of all the *MEN1* germline mutations have been reported as somatic mutations in tumors, suggesting that the mutations likely contribute to clinical pathologies rather than to neoplasia [40]. Interestingly, only a few germline mutations of *MEN1* have been specifically associated to parathyroid tumors, including *P325L* (*P320L* in isoform 2), which is predicted to significantly decrease the protein stability of menin by targeting the protein for degradation [41], and *D423N* (*D418N* in isoform 2), whose molecular consequence is still unknown [40].

### 8.3. Beyond the MEN1 Gene: Lessons from MEN1 Phenocopies

In more than 10% of patients expressing an MEN1 clinical phenotype, no apparent mutation of the MEN1 gene is detected [42]. These cases, defined as MEN1 phenocopies, may depend on a germline mutation of the MEN1 gene in areas not examined by standard genetic assays, such as the 5′ untranslated region, deep introns of the gene, or regulatory regions, or, alternatively, may exhibit MEN1 gene mosaicism. Genetic analysis with next-generation sequencing (NGS) offers new possibilities for detecting these variants [43]. Furthermore, an MEN1-like phenotype may manifest in patients with mutations in other genes, such as the tumor suppressor gene cyclin-dependent kinase inhibitor 1B (CDKN1B). A germinal heterozygote loss-of-function mutation of the CDKN1B gene in a family with an MEN1 phenocopy was first described in 2006 by Pellegata et al. This condition, initialy named as MENX, was later classified as multiple endocrine neoplasia type 4 (MEN4, OMIM #610755) [44]. To date, fewer than 100 cases of MEN4 have been reported in the literature; the typical clinical manifestations are similar to MEN1, including primary hyperparathyroidism, pituitary adenomas, and neuroendocrine tumors; however, they typically present in a milder form and at an older age [45]. Primary hyperparathyroidism is the most prevalent endocrine disorder, occurring in 75–80% of patients, and frequently represents the first clinical manifestation [46]. Interestingly, MEN4-PHPT predominantly occurs from single parathyroid adenomas, whereas diffuse hyperplasia and multiple adenomas are relatively rare. No reported cases of parathyroid carcinomas have been documented to date [47].

The role of CDKN1B in parathyroid tumorigenesis is still a matter of debate. The gene is located on chromosome 12 (12p13) and encodes the protein p27^kip1^, a cyclin-dependent kinase inhibitor that negatively regulates the cell cycle, inhibiting the G1-S phase transition of CDK2 and CDK4 [48]. Deletions or missense mutations of CDKN1B result in a truncated p27 protein, which exhibits a reduced binding capacity, reduced concentration or mislocalization within the nucleus, and reduced stability [49]. As p27^kip1^ functions as a cell cycle regulator, its reduced or compromised activity results in aberrant cell proliferation, promoting tumorigenenesis in various tissues, including parathyroid tissue. Notably, somatic LOH at the CDKN1B locus is rarely observed in MEN4 parathyroid adenomas, although immunohistochemical staining of affected parathyroid tissue does not detect any tissutal expression of the p27^Kip1^ protein. This evidence suggests that CDKN1B does not act as a classic tumor suppressor gene following Knudson’s ‘two-mutation’ criterion and that the down-regulation of p27^Kip1^ protein likely occurs at a post-transcriptional and/or post-translational level [50].

*CDKN1B* mutations represent a potential disease modifier also in patients with *MEN1* mutations. Indeed, menin promotes p27^Kip1^ expression through a transcriptional activation complex that comprises RNA polymerase II (POL III) and methyl-transferase (MLL2). Therefore, the inactivation of menin results in a reduction in the expression of p27 [51]. Mutations in *CDKN1B*, whether germline or somatic, when combined with a mutation in *MEN1*, result in a more significant reduction in the expression of p27 protein, further promoting tumorigenesis [52].

## 9. Menin and Epigenetics

Recent research has also demonstrated that tumorigenesis in MEN1, including development of parathyroid tumors, may be influenced by epigenetic factors. Indeed, menin engages in a multifaceted epigenetic regulation that encompasses a variety of processes, summarized in Table 2.

Menin acts either as an activator or repressor of DNA transcription by interacting with several factors, including the mixed-lineage leukemia (MLL) complex, histone deacetylase (HDAC), or histone lysine methyltransferase SUV39H1. Specifically, menin interacts with MLL, promoting the methylation of the histone H3 (H3K4), which in turn regulates the suppression of the cyclin D kinase inhibitors (CDKi) p18 and p27 and Hox genes, negatively affecting the suppression of cell proliferation [10]. More recently, the loss of menin in MEN1-related tumors was also associated with an increased activity of the DNA (cytosine-5)-methyltransferase 1 (DNMT1), accounting for methylation of cytosine residues of the CpG islands of DNA, thus resulting in silencing gene expression. The hypermethylation of promoters of TSGs following the loss of menin can be regarded as a common pro-oncogenic epigenetic change in MEN1-related pancreatic neuroendocrine tumors [53], and, similarly, it could also occur in *MEN1*-gene-loss-driven parathyroid tumorigenesis. Furthermore, menin was found to interact with HDAC, inducing repression of the transcription factor JunD, involved in negative control of cell proliferation. Moreover, menin is able to silence SUV39H1, a histone 3 lysine 9 (H3K9) methyltransferase regulating both chromatin remodeling and the expression of transcription factors, such as interleukin 6 and gastrulation brain homeobox 2 (Gbx2), involved in cell differentiation [54].

## 10. Haploinsufficiency of Menin and the Role of Micro-RNAs

Menin could have a role also in tissue-specific haploinsufficiency that occurs after the mutation/loss of one copy of the *MEN1* gene, either at the germline or somatic level. In primary cultures of skin fibroblasts obtained from MEN1 patients, mRNAs encoded by the wild-type-retained *MEN1* allele were expressed, while mutant alleles did not, being partially degraded in a nonsense-mediated mRNA decay pathway [55]. This phenomenon results in a reduced expression of *MEN1* mRNA and extremely low levels of menin in MEN1-PHPT with respect to the sporadic PHPT forms and normal parathyroid gland. Furthermore, menin enables the formation of a ribonucleoprotein structure in which multiple mRNAs are coordinately regulated by RNA binding proteins and miRNAs [44]. Such findings suggest that the expression of the *MEN1* gene may be regulated through a mechanism of feedback from its product menin, possibly compensating the haploinsufficiency derived from the allelic loss though an up-regulation of wild-type menin expression at the post-transcriptional level [56].

Many studies have pointed out that also micro-RNA (miRNA) expression may be involved in parathyroid tumorigenesis, unraveling a wide spectrum of differentially expressed miRNAs in parathyroid tumors [57,58,59,60,61]. A study on both healthy and MEN1 parathyroid tissues, with and without LOH at 11q13, has revealed that miR-24-1 may act as an oncomir by blocking menin expression and, consequently, may contribute to parathyroid tumorigenesis via an epigenetic mechanism mimicking the “Knudson’s second hit”. In synthesis, after the first inherited germinal “hit”, the onset and progression of MEN1-associated neoplasms, including those involving parathyroids, could depend on an impairment of the ‘‘negative feedback loop’’ between menin and miR-24-1 [62]. Specifically, MEN1 parathyroid tumors still having the wild-type allele of the *MEN1* gene may exhibit miR-24-1-increased levels in association with the menin complete silencing, thus suggesting a direct miR-24-1-dependent inhibition on *MEN1*-mRNA translation, suggesting that such epigenetic events may drive toward an intermediate molecular step before the genetic LOH of the wild-type *MEN1* allele occurs, consequently inhibiting the menin expression and triggering MEN1 parathyroid tumorigenesis by mimicking the “Knudson’s second inactivating hit” on the *MEN1* gene expression [63]. Therefore, the existence of an autoregulatory network between miR-24-1, *MEN1*-mRNA, and menin has been proposed. Menin promotes the transcription of the primary transcript of miR-24-1 (pri-miR-24-1) by binding the upstream region of the miR-24-1-encoding gene cluster on chromosome 9. Menin directly interacts with pri-miR-24-1, facilitating the DROSHA-mediated processing to pre-miR-24-1, in a positive feedforward loop in which menin promotes the maturation of its own repressor. The mature miR-24-1, together with the RISC complex, binds the 3’ UTR of *MEN1*-mRNA, blocking the translation of menin (Figure 4). In this context, miR-24 may act as a possible effector of tumor development. Thus, the expression of this miRNA only in MEN1 parathyroid adenomas could account for selective parathyroid tumorigenesis, mimicking and substituting the second somatic ‘‘hit’’ of the TSG inactivation, even before the occurrence of LOH [62].

Global miRNA expression profiling in MEN1 parathyroid samples, with and without LOH, identified specific deregulated miRNAs in LOH-positive tumors (LOH^+^) and in LOH-negative tumors (LOH^−^) compared with control samples. The most differentially expressed were miR-1301, miR-664, and miR-4258 [28,64,65]. In silico microarray analyses suggested that miR-1301, as well as miR-664, may potentially target several genes involved in parathyroid tumorigenesis, including *CDKN1B* (Figure 5) [66]. In addition, miR-4258 was reported to be down-regulated in MEN1 LOH^+^ parathyroid adenomas, and it is predicted to target and inhibit the *CCDN1* gene encoding the cyclin D1, which is a positive regulator of the cell cycle and cell growth [28]. Lacking the wild-type menin expression, the parathyroid cell no longer shows the miR-4258-driven negative control of CCND1 expression and, consequently, reveals an increased expression of cyclin D1 and an uncontrolled cell growth.

## 11. Long Non-Coding RNAs (lncRNAs)

A profile analysis of lncRNA expression in parathyroid tumors, including some with MEN1 mutations, showed that the tumors exhibiting *MEN1* gene mutations had an increased expression of six lncRNAs, including BC200, HAR1B, HOXA3as, NEAT1, SNHG6, and ZFAS1. These findings suggest that the *MEN1* gene may also act as a potential modulator of the lncRNA expression. In particular, LncRNA profiling has been suggested as a potential biomarker to distinguish different kinds of parathyroid neoplasms, such as carcinomas, atypical adenomas, and typical adenomas, since these conditions have shown distinct lncRNA profiles. In particular, the expression of BC200 resulted in an increase in parathyroid carcinomas compared to adenomas, suggesting a potential role of BC200 as a novel circulating biomarker for parathyroid carcinomas [66].

## 12. Conclusions

Beyond the realm of genetics, it is essential to consider that also environmental influences contribute significantly to the nuanced manifestation of parathyroid tumors, either at a sporadic or MEN1 level, contributing to the overall variable clinical expressivity of the MEN1 syndrome and, of course, of the MEN1-PHPT. Such variability is observed both at an intrafamilial and interfamilial level, regardless of the type of germline mutation of the *MEN1* gene. As we delve into the intricate world of parathyroid tumorigenesis, it becomes evident that unraveling its mysteries holds promise for not only comprehending the origins of PHPTH but also tailoring precision therapies. The deep exploration of the underlying molecular mechanisms is essential not only for refining diagnostic strategies or precociously identifying subjects at “hereditary” risk of developing PHPT, but also for advancing targeted interventions that may mitigate the impact of parathyroid tumors on bone health, renal function, and overall well-being.

This journey into the heart of parathyroid tumorigenesis associated with MEN1 syndrome invites us to navigate the intricate molecular landscapes, decode the genetic signatures, and appreciate the dynamic interplay between inherent susceptibilities and external triggers. In doing so, we embark on a path toward a deeper understanding of this endocrine disorder, laying the groundwork for innovative approaches to diagnosis, treatment, and, perhaps one day, prevention.

## Figures and Tables

**Figure 1 ijms-25-11586-f001:**
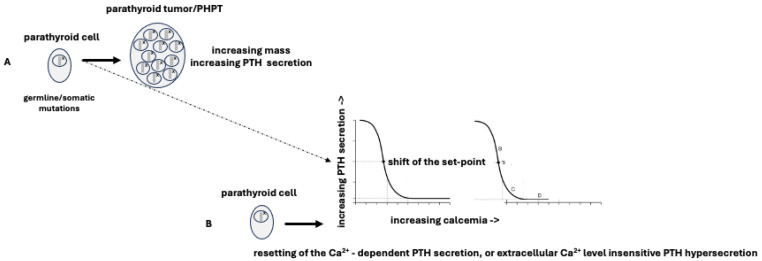
Simplified summary of parathyroid tumorigenesis/increasing PTH secretion. (**A**) Increasing mass of the parathyroid cells due to mutations (small x) determining a “proliferative disturbance”; (**B**) Resetting of the Ca^2+^-dependent PTH secretion, or extracellular Ca^2+^ level-insensitive PTH hypersecretion due to a “functional disturbance” of the PTH set-point due to mutations (small x) disrupting the normal pathways of PTH secretion. The dotted arrow indicates the rather robust possibility that, in addition to deranging of cell growth, deranging of Ca^2+^-dependent PTH secretion may also coexist in parathyroid tumors.

**Figure 2 ijms-25-11586-f002:**
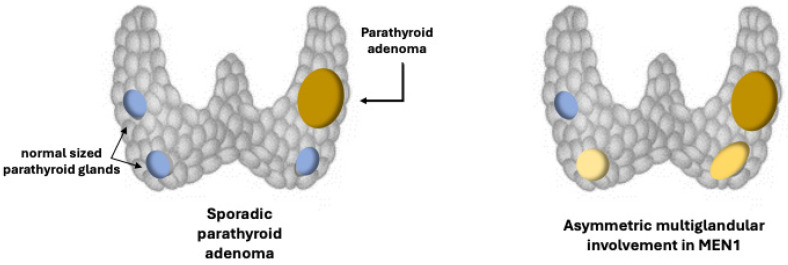
Asymmetric (and asynchronous) growth of the affected parathyroid gland in MEN1. Such an abnormal growth mode may explain why it can be possible to miss the finding of a multiglandular involvement at the time of initial surgery.

**Figure 3 ijms-25-11586-f003:**
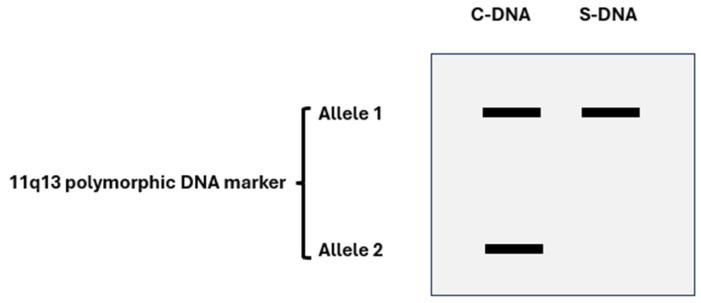
Schematic and generic exemplification of LOH, as it can be observed at a gel electrophoresis analysis of polymerase chain reaction (PCR)-generated DNA fragments (polymorphic DNA marker). A given polymorphic DNA marker, at a given chromosome locus, provides the opportunity to detect two allelic variant s (allele 1 and allele 2, heterozygosity). C-DNA (constitutive DNA) retains both the alleles, while S-DNA (somatic DNA, e.g., from a tumor DNA sample) does not (loss of allele 2, or LOH).

**Figure 4 ijms-25-11586-f004:**
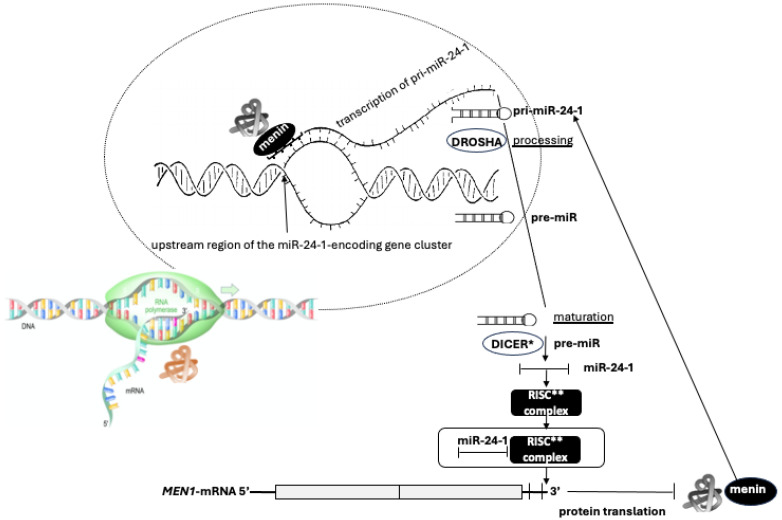
Extreme simplification of the possible autoregulatory interacting network between miR-24-1, *MEN1*-mRNA, and menin. In the upper part of the figure, menin promotes the transcription of pri-miR-24-1 by binding to the upstream region of the miR-24-1-encoding gene cluster. So, it directly interacts with pri-miR-24-1, representing the primary transcript of that miR, and then, through DROSHA (a class 2 ribonuclease III enzyme) cooperation, favors the processing to pre-miR-24-1. Both processes occur internally in the nuclear envelope, represented here by the fine dashed line. Then, in the cytoplasm, lower part of the figure, menin promotes the maturation to pre-miR and miR-24-1, which represents the repressor of its own expression. Finally, miR-24-1, interacting with ** RISC (RNA-induced silencing complex), binds the 3′ untranslated region (UTR) of *MEN1*-mRNA blocking menin’s translation. * DICER = RNase III endonuclease that processes miRNA precursors.

**Figure 5 ijms-25-11586-f005:**
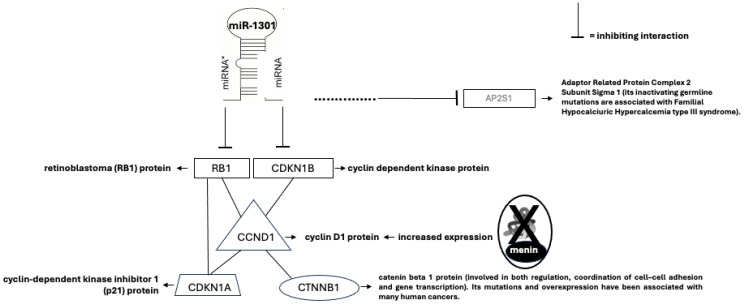
miR-1301 interacts with retinoblastoma (RB1), an onco-suppressor protein, and with cyclin-dependent kinase inhibitor 1B (CDKN1B) (p27^Kip1^) in MEN1 parathyroid tumorigenesis. RB1 protein blocks the cell cycle progression, thus preventing/limiting incorrect or harmful divisions. Inactivating mutations of RB1 gene have been reported in several types of human tumors. CDKN1B gene encodes a protein belonging to a family of cyclin-dependent kinase (Cdk) inhibitor proteins. This protein binds to and prevents the activation of cyclin E-CDK2 or cyclin D-CDK4 complexes, thus controlling the cell cycle progression at G1 stage. Germline mutations of CDKN1B gene have been reported to be causal of MEN4 syndrome, in which PHPT may frequently occur. Molecular interactions between the pathways of the RB1 protein and those underlying cyclin D kinase inhibitors have been described in the literature, and the pharmacological inhibition of cyclin D kinase inhibitors represents an important approach to provide new therapeutic strategies in the oncology field, and not only for the interaction with the RB1 pathways. In the ellipse, bottom right, it is indicated that the lack of expression of menin, X, determines the loss of the miR-4258-driven negative control of CCND1 expression and, consequently, an increased expression of cyclin D1. * is for miRNA = microRNA.

**Table 1 ijms-25-11586-t001:** Familial forms of PHPT, syndromic and non-syndromic, causal genes, chromosomal loci, and heredity.

Disorder	Gene	Chromosomal Locus	Heredity
Syndromic			
MEN1	*MEN1*	11q13.1	AD
MEN2A	*RET*	10q11.21	AD
MEN4	*CDKN1B*	12p13.1	AD
HPT-JT	*CDC73* (*HRPT2*)	1q31.2	AD
Non-syndromic			
FIHP	*MEN1*, *CDC73*, *CaSR* (and others still unknown)	11q13.1, 1q31.2, and 3q13.3–q21.1	AD
FHH type 1	*CaSR*	3q13.3–q21.1	AD
FHH type 2	*GNA11*	19p13.3	AD
FHH type 3	*AP2S1*	19q13.32	AD
NSHPT	*CaSR*	3q13.3–q21.1	AR

Remember that the syndromic forms of FPHPT represent 2–5% (in any case < 10%) of overall PHPT. MEN = multiple endocrine neoplasia syndrome; FIHP = familial isolated hyperparathyroidism; FHH = familial hypercalcemia hypocalciuria; NSHPT = neonatal severe hyperparathyroidism; AD = autosomal dominant; AR = autosomal recessive.

**Table 2 ijms-25-11586-t002:** Some of the epigenetic functions associated with the menin protein are summarized.

Epigenetic Function	Description
Histone Modification	Menin interacts with histone-modifying complexes (e.g., MLL1/2) to regulate histone methylation status.
Chromatin Remodeling	Menin participates in chromatin-remodeling complexes, influencing chromatin structure and gene expression.
Transcription Regulation	Menin modulates gene transcription by interacting with transcription factors and regulating their activity.
DNA Repair	Menin plays a role in DNA damage processes, impacting genome stability and integrity.
Cell Cycle Regulation	Menin contributes to cell cycle control by regulating various genes involved in cell division.

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
