# Peer review of "Molecular Pathophysiology of Parathyroid Tumorigenesis—The Lesson from a Rare Disease: The “MEN1 Model”"

_ijms, 2024, doi:10.3390/ijms252111586_

Round 1

Reviewer 1 Report

Comments and Suggestions for Authors

This article does a comprehensive review of MEN1 from clinical manifestations, to molecular mechanisms.  It is in-depth, and well written.  A few minor comments to strengthen the papers.

1.  A few areas of the paper have some lay terminology such as "fascinating topic" and "captivating puzzle."  Given the scientific terminology in the rest of the manuscript the authors should consider removing lay terminology throughout.  Also highly penetrant (not penetrating).

2.  The nomenclature of MEN1 should be consistent.  MEN1 (italics) is the gene, menin is the protein, and MEN1 (no italics is the syndrome).  Other terms used are MEN-1, MEN(space)1 etc.

3. The sentence: "Several germline or somatic mutations of the MEN1 gene have been described to date both in inherited and de novo cases." is clunky.  Germline mutations and somatic mutations are usually kept separate.  Germline are mutations that are the first hit and can be passed down to offspring.  Somatic are in non-hereditary cases, or the second hit in germline cases.

4. A discussion of MEN4 should be incorporated into the overall paper.  This can be incorporated into the signaling, or a brief paragraph on it.

Comments on the Quality of English Language

see above in minor comments.

Author Response

Reviewer 1: This article does a comprehensive review of MEN1 from clinical manifestations, to molecular mechanisms.  It is in-depth, and well written.  A few minor comments to strengthen the papers.

1.  A few areas of the paper have some lay terminology such as "fascinating topic" and "captivating puzzle."  Given the scientific terminology in the rest of the manuscript the authors should consider removing lay terminology throughout.  Also highly penetrant (not penetrating).

2.  The nomenclature of MEN1 should be consistent.  MEN1 (italics) is the gene, menin is the protein, and MEN1 (no italics is the syndrome).  Other terms used are MEN-1, MEN(space)1 etc.

3. The sentence: "Several germline or somatic mutations of the MEN1 gene have been described to date both in inherited and de novo cases." is clunky.  Germline mutations and somatic mutations are usually kept separate.  Germline are mutations that are the first hit and can be passed down to offspring.  Somatic are in non-hereditary cases, or the second hit in germline cases.

4. A discussion of MEN4 should be incorporated into the overall paper.  This can be incorporated into the signaling, or a brief paragraph on it.

Response: 

Reply to Reviewer 1

We appreciate the reviewer's constructive comments.  The text has been revised in accordance with the suggestions. (The corrections are marked in red). Specifically:

1-3. The terminology has been changed, highlighting the differences between MEN1 (syndrome) and MEN1 (gene).  The sentence about germline/somatic mutations was corrected.

4. A new section discussing MEN4 and the function of CDKN1b in parathyroid tumor development has been incorporated into the text, along with the corresponding references.

Reply to Reviewer 2
Comments 2: This manuscript is a mini review on MEN1-associated parathyroid tumorigenesis, based on studies retrieved from PubMed.
The manuscript is very well written and presented.
The sequence of pathophysiological events associated with genetic mutation, at germline and/or somatic level promoting neoplastic transformation of parathyroid cells is highlighted.
In addition, the genetic/molecular studies of parathyroid tumorigenesis associated with MEN1  Syndrome that represent a model of understanding genetic-epigenetic-molecular aspects underlying the pathophysiology of primary hyperparathyroidism are emphasized.
Finally, clinical, epidemiological aspects and specific guidelines based on molecular-genetics studies and reviews are considered.
Response 2: We are deeply grateful to the reviewer for their considerations and for the positive comments on our manuscript.

Reviewer 2 Report

Comments and Suggestions for Authors

This manuscript is a mini review on MEN1-associated parathyroid tumorigenesis, based on studies retrieved from PubMed.

The manuscript is very well written and presented.

The sequence of pathophysiological events associated with genetic mutation, at germline and/or somatic level promoting neoplastic transformation of parathyroid cells is highlighted.

In addition, the genetic/molecular studies of parathyroid tumorigenesis associated with MEN1  Syndrome that represent a model of understanding genetic-epigenetic-molecular aspects underlying the pathophysiology of primary hyperparathyroidism are emphasized.

Finally, clinical, epidemiological aspects and specific guidelines based on molecular-genetics studies and reviews are considered.

Author Response

Reply to Reviewer 2

Comments 2: This manuscript is a mini review on MEN1-associated parathyroid tumorigenesis, based on studies retrieved from PubMed.

The manuscript is very well written and presented.

The sequence of pathophysiological events associated with genetic mutation, at germline and/or somatic level promoting neoplastic transformation of parathyroid cells is highlighted.

In addition, the genetic/molecular studies of parathyroid tumorigenesis associated with MEN1  Syndrome that represent a model of understanding genetic-epigenetic-molecular aspects underlying the pathophysiology of primary hyperparathyroidism are emphasized.

Finally, clinical, epidemiological aspects and specific guidelines based on molecular-genetics studies and reviews are considered.

Response 2: We are deeply grateful to the reviewer for their considerations and for the positive comments on our manuscript.
